# Is the Life History Flexibility of Cold Desert Annuals Broad Enough to Cope with Predicted Climate Change? The Case of *Erodium oxyrhinchum* in Central Asia

**DOI:** 10.3390/biology10080780

**Published:** 2021-08-16

**Authors:** Huiliang Liu, Yanfeng Chen, Lingwei Zhang, Jerry M. Baskin, Carol C. Baskin, Lan Zhang, Yan Liu, Daoyuan Zhang, Yuanming Zhang

**Affiliations:** 1State Key Laboratory of Desert and Oasis Ecology, Xinjiang Institute of Ecology and Geography, Chinese Academy of Sciences, Urümqi 830011, China; liuhuiliang@ms.xjb.ac.cn (H.L.); chenyanfeng@qfnu.edu.cn (Y.C.); zhangdy@ms.xjb.ac.cn (D.Z.); 2Yili Botanical Garden, Xinjiang Institute of Ecology and Geography, Xinyuan 835800, China; 3School of Geography and Tourism, Qufu Normal University, Rizhao 276800, China; lathyrus@163.com; 4Xinjiang Key Laboratory of Soil and Plant Ecological Processes, College of Life Sciences, Xinjiang Agricultural University, Urümqi 830052, China; zlwlz@163.com (L.Z.); z13150306508@163.com (L.Z.); 5Department of Biology, University of Kentucky, Lexington, KY 40506, USA; Jerry.baskin@yahoo.com (J.M.B.); carol.baskin@uky.edu (C.C.B.); 6Department of Plant and Soil Sciences, University of Kentucky, Lexington, KY 40506, USA; 7Turpan Eremophytes Botanical Garden, Chinese Academy of Sciences, Turpan 838008, China

**Keywords:** annual plant, climate change, dry spring/autumn, *Erodium oxyrhinchum*, F2 seed dormancy, life history flexibility, wet spring/autumn

## Abstract

**Simple Summary:**

The broad objective of our study was to evaluate the life history flexibility of the annual *Erodium oxyrhinchum* in relation to the predicted changes in precipitation in the cold desert due to climate change. We compared the effects of dry and wet springs and dry and wet autumns on growth and F2 seed dormancy of plants from spring- and autumn-germinated seeds of the cold desert annual *E. oxyrhinchum* from 2016–2020. Also, the future climate for the region was modeled. Our results found that given the flexibility in the life history of *E. oxyrhinchum*, it seems unlikely that future climate change will have much of an impact on the life history and dominant position of this species in the herbaceous plant community in the unpredictable-rainfall environment of the cold desert. Our study challenges the commonly-held perception that desert organisms may be negatively affected by climate change.

**Abstract:**

Interannual seasonal variability in precipitation may strongly affect the life history and growth of desert annual plants. We compared the effects of dry and wet springs and dry and wet autumns on growth and F2 seed dormancy of plants from spring (SG)- and autumn (AG)-germinated seeds of the cold desert annual *Erodium oxyrhinchum*. Vegetative and reproductive growth and F2 seed dormancy and germination were monitored from September 2016 to November 2020 in the sandy Gurbantunggut Desert in NW China in Central Asia. Dry autumns decreased the density of AG plants, and dry springs decreased the density of SG plants and growth of SG and AG plants. In dry springs, SG plants were more sensitive to precipitation than AG plants, while in wet springs SG and AG plants had similar responses to precipitation. During growth in both dry and wet springs, most morphological characters of SG and AG plants initially increased rapidly in size/number and then plateaued or decreased, except for SG plants in dry springs. In dry springs, most morphological characters of AG plants were larger or more numerous than those of SG plants, and they were larger/more numerous for SG plants in wet than in dry springs. The percentage biomass allocated to reproduction in SG plants was slightly higher in a wet than in a dry spring. A much higher proportion of dormant seeds was produced by AG plants in a wet spring than in a dry spring. Projected changes in precipitation due to climate change in NW China are not likely to have much of an effect on the biology of this common desert annual plant.

## 1. Introduction

Water is the main limiting factor for plant growth and development in deserts [1,2,3,4,5,6], and thus the responses of desert plants are mainly regulated by the availability of water [4,5,7]. Seed germination and seedling survival are the most sensitive stages of plant reproduction, and they are strongly affected by available soil moisture [4,5,6,8]. One of the questions with regard to global climate change is: how will the amount and timing of precipitation be affected by it? This is a critical question for annual plants that grow in arid regions, such as cold deserts, where timing of the unpredictable precipitation greatly influences seed germination, seedling survival, and seed production by mature plants [9,10,11,12].

The precipitation in central Asia also shows obvious interannual variation [13], whereas the annual precipitation is falling mainly on winter and spring (accounting for up to 80% of the annual total precipitation) in the southern regions [14]. The Gurbantunggut Desert, a cold sandy desert in Xinjiang Province in northwestern China (Central Asia), is the largest fixed or semi-fixed sand desert in China and covers 4.88 × 10^4^ km^2^ [15]. In a typical year, precipitation in this desert is about 180 mm (rainfall and snow), and most of it occurs in spring and summer. The interannual precipitation pattern in NW China is much more dramatic than it is in the other deserts in China [16,17,18]. By analyzing precipitation data from the Fukang meteorological station in the Gurbantunggut Desert for the last 55 years, we found that interannual precipitation differed significantly, with the highest amount of precipitation (388.6 mm) in 1996 and the lowest (77.7 mm) in 2020, a five-fold difference. The coefficient of variation (cv) in amount of precipitation for years, spring, summer, autumn, and winter is 28%, 44.8%, 46.1%, 40.9%, and 45.8%, respectively. Thus, wet vs. dry seasons are defined as whether actual precipitation values are higher or lower than the mean precipitation values in the different seasons.

This strong interannual seasonal variation in precipitation affects many aspects of plant growth [10,11,19]. Due to interannual variation in precipitation, seeds of annual/ephemeral species in the Gurbantunggut Desert, such as *Diptychocarpus strictus* [20], *Eremopyrum distans* [9,21], *Erodium oxyrhinchum* [22,23], and *Isatis violascens* [24], may germinate only in spring or in spring and autumn, depending on when the soil is sufficiently moist for them to do so [25,26]. However, it can be too dry even in spring for germination to occur [5,8,11]. For example, seeds of the four cold desert species *Alyssum desertorum*, *Lappula semiglabra*, *Plantago minuta,* and *Tetracme recurvata* did not germinate when soil water content in a dry spring was <3%, whereas germination was 60–80% in a wet spring [27]. When normal snow cover depth (>20 cm) was doubled artificially, seedling density of desert annual plants was almost 10 times higher than it was in a year with normal snow cover due to germination in spring [28].

Climate change models for NW China predict that the precipitation patterns will change. Spring and summer are predicted to be wetter and autumn and winter drier than at present [18,29,30]. In addition, extreme precipitation events will increase significantly [18,30]. The models also predict that the present trend of increasing precipitation in the cold deserts of Central Asia will continue in the 21st century [17,18]. The eastern part of the cold desert is predicted to become wetter (Tianshan Mountains) and the western part drier, allowing the most arid area of the cold desert to expand eastward [16,28].

The broad objective of our study was to evaluate the flexibility of the life history of the cold desert annual *Erodium oxyrhinchum* in relation to the predicted changes in precipitation in the cold desert due to climate change. We hypothesized that life history flexibility of this biseasonal-germinating species is broad enough for populations to persist in situ in the cold desert of Central Asia into the next century. This study involved: (1) a detailed consideration of predicted changes in precipitation and temperature from climate change models for NW China; (2) monitoring temperature, precipitation, and soil moisture in the field for 4 years; (3) monitoring growth of plants that naturally established in the field in autumn and in spring in relation to amount of available soil moisture; and (4) determination of F2 seed dormancy. We predicted that four consecutive years of monitoring would capture the normal range of variability in precipitation predicted by climate change models, thus providing insight into the persistence (or not) of *E. oxyrhinchum* populations in the Central Asian cold desert into the future.

*Erodium oxyrhinchum* is the most common annual ephemeral plant in early spring in the Gurbantunggut Desert [31,32,33,34], and it often germinates only in spring. It is cross-pollination, relying on insects like bees. *E. oxyrhinchum* seeds have physical dormancy, and dormancy release needed high temperature in summer and low temperature in autumn and winter. However, seeds can germinate in autumn if the soil is moist [12,35,36]. Spring-germinated (SG) plants complete their ephemeral life cycle in about 2 months. Autumn- germinated (AG) plants overwinter as rosettes and complete their (winter annual) life cycle the following spring [22,36]. The seeds from both SG and AG plants of *E. oxyrhinchum* have physical dormancy (PY), but the percentage of seeds with PY (vs. those with nondormancy) differed significantly [37]. In addition, under increasing precipitation or precipitation plus nitrogen, growth, development, and reproduction of SG plants were more sensitive than AG plants [12,35].

## 2. Materials and Methods

### 2.1. Study Site

The study was conducted from 1 September 2016 to 1 November 2020 at a site on the southern margin of the Gurbantunggut Desert (44°26′ N, 87°54′ E, 532 m a.s.l.). Mean annual air temperature is 6.6 °C, and the extreme high and low air temperatures are 42.6 (July) and −41.6 °C (January), respectively. Mean annual monthly precipitation varies from 70 to 180 mm, and mean annual evaporation is about 2000 mm [38]. The dominant plants at the study site are the shrubs *Haloxylon ammodendron* and *H. persicum*, which cover about 30% of this desert [39]. Common annual ephemerals include *Agriophyllum squarrosum*, *Alyssum linifolium*, *E. oxyrhinchum*, *Lactuca undulata*, *Nepeta micrantha* and *Schismus arabicus*, and the common perennial ephemerals are *Carex physodes*, *Eremurus inderiensis*, *Euphorbia turczaninowii,* and *Orostachys spinosa* [15]. Our study was done in a relatively homogeneous flat interdune swale, where the density of naturally-germinating *E. oxyrhinchum* plants can reach 96.5 plants/m^2^ [12,35].

### 2.2. Climate Data for the Last Half Century in the Gurbantunggut Desert

We used monthly precipitation data from the nearest weather station (Fukang) collected between 1965 and 2020. We divided the amount of precipitation into spring (March, April and May), summer (June, July, and August), autumn (September, October, and November), and winter (December, January, and February).

### 2.3. Predictions of Climate Change Models

We obtained high-resolution precipitation data for our field sites from the Coupled Model Intercomparison Project 6 (CMIP6) climate projections data, which consists of information (“runs”) from about 100 climate models produced from 49 different modeling groups. These data were downloaded from the Climate Data Store [40]. We chose three Shared Socioeconomic Pathway (SSP) scenarios: SSP1, SSP2, and SSP5 with greenhouse gas emissions that potentially would result in a 1.9, 4.5, and 8.5 °C increase in global temperature, respectively, by 2100. SSP1, SSP2, and SSP5 represent economic and social development path, moderate development path, and conventional development path, respectively [41]. In addition, we used the CAMS-CSM1-0 model to project changes in precipitation. The model output consists of monthly precipitation in the future (to 2099). We divided the amount of precipitation into that of spring, summer, autumn, and winter, as defined above.

### 2.4. Precipitation, Temperature, and Soil Water Content

On 1 September 2016, we installed a weather monitoring device (Caipos GmbH, Schillerstrasse Gleisdorf, Austria) at the study site to measure amount of precipitation and soil temperature and soil moisture content, which were recorded hourly at a depth of 5 cm.

### 2.5. Emergence and Density of AG and SG Plants

As newly-germinated SG and AG seedlings emerged, they were marked by placing a red and white wire ring, respectively, around the stem. SG and AG plants were assigned to a 1 m × 1 m subplot within each of nine 3 m × 5 m plots for biomass sampling. From 10 October 2016 to 10 October 2020, the density of AG plants was recorded annually, and 16 AG plants were randomly marked in each of the nine subplots. From 10 March 2017 to 20 April 2020, the density of SG plants was recorded annually, and 16 SG plants were randomly marked in each of the nine subplots. We designated seeds that developed into SG and AG plants as the F1 generation and seeds produced by SG and AG plants as the F2 generation.

### 2.6. Morphological Characters of SG and AG Plants

SG and AG plants were sampled every 8 days during the four springs growing season (20 March to 20 May) of 2017 to 2020. We measured/counted plant height, root length, number of leaves, leaf area and number of branches for SG and AG plants. For leaf area, we collected all leaves from each plant and measured total area with a LI-COR 3000 leaf area meter.

### 2.7. Dry Mass Accumulation and Allocation in SG and AG Plants

For SG and AG plants in 2017 to 2020, we divided plants into roots (washed free of soil), stems, leaves and reproductive organs (flowers, fruits, and seeds). We harvested seeds from SG and AG plants when fruits were dry, yellow, and dehiscing. Fruits, leaves, stems, and roots (washed free of soil) of each plant were weighed using a Sartorius BS210S electronic-balance (0.0001 g) after drying them at 75 °C for 48 h. Total biomass is the sum of roots, stems, leaves, and reproductive organs per plant. Allocation of biomass was determined as described by Chen et al. [12,35].

### 2.8. Offspring (F2) Seed Dormancy and Germination of AG and SG Plants

We collected mature fruits from SG and AG plants from 2017 to 2020 and removed the seeds from them. Germination tests were conducted on the freshly-matured seeds at 25/10 °C (12/12 h, light/dark). This temperature regime simulates the mean daily maximum and minimum air temperatures in early spring in the Gurbantunggut Desert. Twenty-five seeds were placed in each of four 7-cm-diameter Petri dishes on two layers of Whatman No. 1 filter paper moistened with 3 mL of distilled water. Water was added as needed to keep the filter paper moist during the germination tests. Germinated seeds (emerged radicle) were counted and removed from the Petri dishes daily for 30 days, and final germination percentage (FGP) was determined as described by Chen et al. [12,33]: FGP = Germinated seeds (GN)/ Total number of viable seeds tested (SN). After the germination experiment ended, viability of nongerminated seeds was determined using TTC (2, 3, 5-Triphenyltetrazolium chloride) [42].

### 2.9. Statistical Analyses

Data were arcsine (percentage data) or log10 (other data) transformed before analysis to approximate normal distribution and homogeneity of variance. If the variance of the transformed data still did not meet the requirement of normal distribution, we assessed the differences between treatments by the Kruskal–Wallis non-parametric test. Tukey’s test for multiple comparisons was used to determine significant differences among treatments. To avoid type I error problems, a Bonferroni correction was performed. Plant height, root length, number of leaves, leaf area, number of branches, individual biomass, and proportion of biomass allocated to roots, stems, leaves, and reproductive organs and F2 seed dormancy were analyzed as dependent variables with one-way ANOVA. We considered germination times (AG plants in 2016, 2018, and 2019, SG plants in 2017, 2018, 2019, and 2020) as fixed effects. Data were analyzed using the software SPSS 13.0 (SPSS Inc., Chicago, IL, USA). All figures were drawn with Origin software 2015 (Origin Lab, Northampton, MA, USA).

## 3. Results

### 3.1. Climate Data for the Last Half Century in the Gurbantunggut Desert

Precipitation differed significantly between years. The highest amount of precipitation occurred in 1996 (388.6 mm) and the lowest amount in 2020 (77.7 mm), a five-fold difference. Precipitation also differed significantly between seasons. The highest amount of precipitation in spring was 166.7 mm (1998) and the lowest amount 11.6 mm (1965), a >14-fold difference. The highest amount of precipitation in autumn was 106.6 mm (1974) and the lowest amount 14.4 mm (2020), a >seven-fold difference (Figure 1).

### 3.2. Predictions from Climate Change Models

In the SSP1 1.9 scenario, the CAMS-CSM1-0 model shows that the difference in the amount of interannual precipitation will decrease slightly from 2021 to 2099. In the SSP2 4.5 and SSP5 8.5 scenarios, the model shows that interannual precipitation will increase significantly from 2021 to 2099 (Figure 2). However, in all scenarios, the fluctuation and trend of seasonal precipitation will be significantly different in the future than it is at present. In the SSP1 1.9 scenario, precipitation will decrease slightly in spring and autumn and increase significantly in summer and winter (Figure 3). In the SSP2 4.5 and SSP5 8.5 scenarios, the amount of precipitation will increase significantly in all seasons (Figure 3).

### 3.3. Precipitation and Temperature in Autumn and Spring

Average daily soil temperature and daily precipitation at the study site from 1 September 2016 to 1 November 2020 are shown in Figure 4a. The temperature reached a minimum of −33.8 °C on 27 January 2018 and a maximum of 39.2 °C on 11 July 2019. From September to November 2016, 2018, and 2019, there were 4, 3, and 5 precipitation events of >5 mm, respectively, which provided favorable soil moisture for the emergence and growth of AG plants (Figure 4b). However, from September to November 2017 and 2020 there were only one and no precipitation events of >5 mm, respectively. Amount of precipitation in spring 2017 (89.3 mm) and 2018 (84.1 mm) was significantly higher than that in spring 2019 (35.7 mm) and 2020 (23.1 mm) (Figure 4a). Snow depth in winter 2016, 2017, 2018, and 2019 was 36.7, 28.7, 27.7, and 33.0 cm, respectively. Water from melting snow in early spring and precipitation in the late spring and summer was relatively abundant in 2017 and 2018. Thus, autumns of 2016, 2018 and 2019 were defined as wet and autumns of 2017 and 2020 as dry, springs of 2017 and 2018 as wet, and springs of 2019 and 2020 as dry.

### 3.4. Emergence and Density of AG and SG plants

The wet autumns of 2016 (58.4 mm), 2018 (81.0 mm), and 2019 (64.9 mm) promoted seedling emergence (AG plants), and the density was 20, 18, and 46 seedlings per m^2^, respectively. Water from melting snow promoted seedling emergence (SG plants) in early spring, and the density in 2017, 2018, 2019, and 2020 was 33, 43, 14, and 6 seedlings per m^2^, respectively. Thus, in spring 2017, 2019, and 2020, both AG and SG plants of *E. oxyrhinchum* were present in the population. Precipitation was infrequent from late spring to autumn 2017 (49.4 mm) and 2020 (14.4 mm), and no seeds of *E. oxyrhinchum* germinated in autumn (Figure 4c). Thus, the amount and time of precipitation in autumn (dry vs. wet) affected the emergence and density of AG plants, and that in spring (dry vs. wet) not only affected the emergence and density of SG plants but also the growth of both AG and SG plants.

### 3.5. Morphological Characters of SG and AG Plants

For SG and AG plants in wet springs and for AG plants in dry springs, growth was rapid during the vegetative stage, but then it tended to plateau or decrease during the reproductive stage; For SG plants in dry springs, growth was very slow during the vegetative and reproductive stages (Figure 5, Figure 6, Figure 7, Figure 8 and Figure 9). In dry springs, all morphological characters of AG plants were larger in size/higher in number than those of SG plants. In wet springs (2017 and 2018), the root length of SG plants did not differ significantly from that of SG plants in dry springs (2019 and 2020), while root length of SG and AG plants in 2017 was significantly higher than that of SG plants in 2018 (Figure 6). Conversely, the leaf area of SG plants in wet springs was significantly greater than that of SG plants in dry springs, while the initial leaf area, number of leaves, and number of branches of AG plants in dry springs were higher than for SG and AG plants in wet springs (Figure 7, Figure 8 and Figure 9).

### 3.6. Dry Mass Accumulation and Allocation in SG and AG Plants

There was a significant positive correlation between morphological characters (height, root length, number of leaves, leaf area, and number of branches) and biomass accumulation of plants of *E. oxyrhinchum* (Appendix A). Biomass accumulation of both SG and AG plants in wet springs but only that of AG plants in dry springs increased rapidly during the vegetative stage and then tended to level-off during the reproductive stage. Biomass accumulation of AG plants in dry and wet springs and of SG plants in wet springs was significantly greater than that of SG plants in dry springs (Figure 10).

During the vegetative stage, roots, stems, and leaves of both SG and AG plants in dry and wet springs accounted for all biomass allocation, and allocation of biomass to leaves was greater than that to roots and stems (Figure 11). As plants matured, allocation of biomass to leaves and roots decreased gradually, allocation to stems increased gradually, and allocation to reproductive parts began (Figure 11). In the late growth stage in wet springs, biomass allocation to leaves of both SG and AG plants began to decrease rapidly, and that to reproductive organs began to increase rapidly. During the reproductive stage, allocation of biomass to stems and reproductive organs was higher than that to roots and leaves. In dry springs, biomass allocation in SG plants differed from that in SG plants in wet springs. In the late growth stage, allocation of biomass to leaves and roots was higher than that to stems and reproductive organs (Figure 11).

### 3.7. F2 Seed Dormancy and Germination of AG and SG Plants

The number of seeds produced was higher for both SG and AG plants in wet springs than in dry springs, except for AG plants in the dry spring of 2019 (Figure 12). The percentage of germinated, dormant, and nonviable seeds differed between SG and AG plants in both wet and dry springs. A higher proportion of dormant seeds of AG plants was produced in wet than in dry springs, while proportion of dormant seeds of SG plants produced in wet vs. dry springs did not differ significantly (Figure 13). The percentage of nonviable F2 seeds produced by SG plants in wet springs was 32.0 (2017) and 2.0 (2018) and in dry springs 1.0 (2019) and 100 (2020). The percentage of nonviable F2 seeds produced by AG plants in wet springs was 7.0 (2016) and in dry spring 28.0 (2018) and 12.7 (2019). Dormancy of F2 seeds produced by SG plants in the wet springs was 35.0% (2017) and 96.0% (2018) and in dry springs was 77.0% (2019) and 0% (2020). However, the dormancy of F2 seeds produced by AG plants in wet springs was 84.0% (2016) and 27.0% (2018) and in the dry spring 37.6% (2019). Germination of F2 seeds produced by SG plants in wet springs was 33.0% (2017) and 2.0% (2018) and in dry springs was 22% (2019) and 0% (2020). However, germination of F2 seeds produced by AG plants in wet springs was 9.0% (2016) and 45.0% (2018) and in the dry springs was 49.7% (2019) (Figure 13).

## 4. Discussion

Over the past half century, yearly precipitation has been highly variable in our study area in NW China. There were six, three, and five precipitation events of >5 mm in the study area from September to November 2016, 2018 and 2019, respectively, and the total precipitation for this period was 85.83, 81.0, and 64.9 mm, respectively (Figure 4a), which stimulated emergence of AG plants in late autumn 2016, 2018, and 2019. Thus, the population of *E. oxyrhinchum* consisted of both AG and SG plants in the wet spring of 2017 and the dry springs of 2019 and 2020. Conversely, there were three (all in November 2017) and zero precipitation events of >5 mm in the study area from September to November in 2017 and 2020, respectively (Figure 4a), and total amount of precipitation in autumn 2017 and 2020 was 38.3 and 14.4 mm, respectively. Further, when the soil was moist the average daily temperature had decreased to below 0 °C (2017) or the soil was dry (2020). Thus, in autumn 2017 and 2020, no AG seedlings of *E. oxyrhinchum* emerged. Therefore, temperature is the main environmental factor driving the emergence of SG plants, but precipitation has an important influence on the emergence of AG plants.

In both wet and dry springs, precipitation and/or water from snowmelt and favorable temperatures promoted germination. Thus, the population of *E. oxyrhinchum* consisted of both AG and SG plants in both wet and dry springs, except in the spring of 2018 (wet spring) when AG plants were absent. Clearly, then, emergence of SG and AG plants was determined by soil moisture in both spring and autumn. Spring is a predictable time for seed germination of annual plants of the cold desert of NW China, while autumn is unpredictable. Previous studies have shown that seed germination of annual ephemeral plants depends on the soil moisture when temperatures are favorable for germination [8,9,42], such as in early spring, autumn, and even late summer [12,24,43]. Thus, small precipitation events were unable to trigger germination for *E. oxyrhinchum* in dry autumn, it may be a common phenomenon in desert ecosystems [44,45]. Lu et al. (2014) also found that biseasonal (autumn and spring) germination of the annual *Diptychocarpus strictus* is an adaptation to the variation in timing and amount of rainfall in its rainfall-unpredictable habitat in the cold deserts of Central Asia [20]. In addition, other results that showed water deficit also delayed seedling emergence and led to a low survival rate [45]. The precipitation event and length of interval between two precipitation events affected seed germination and seedling emergence [46]. Our result showed that seeds can germinate and survive at intervals of precipitation events in both dry and wet springs.

Most morphological characters, including height, root length, number of leaves, leaf area, and number of branches during the vegetative growth stage of both SG and AG plants of *E. oxyrhinchum* in wet springs and of AG plants in dry springs increased rapidly in size/number, most likely due to the relatively abundant precipitation and snowmelt water in early spring (Figure 5, Figure 6, Figure 7, Figure 8 and Figure 9). In addition, initial (when plants were marked) leaf area, number of leaves and number of branches of AG plants in dry springs (2019 and 2020) was higher than that of SG and AG plants in wet springs (2017 and 2018). The likely reason for the differences is that AG plants germinated and grew rapidly in wet autumns (2018 and 2019), and they quickly resumed growth in spring. Soil (sand) moisture content in early spring was >6% for about 1 month after seed germination in both dry and wet springs (Figure 4b). Therefore, early spring was suitable for growth of SG and AG plants. SG and AG plants set seeds in both wet and dry springs, except for SG plants in the dry spring of 2020, and they died as the soil moisture content decreased and temperatures increased in late spring and early summer. All morphological characters of AG plants were greater in size/number than those of SG plants in both dry and wet springs, which likely was due to AG plants having a longer life cycle and thus attaining a larger size than SG plants, as has been shown for other Central Asian cold desert annual species with both SG and AG plants [12,42]. Furthermore, most morphological characters of SG plants in wet springs were larger and more numerous than those of SG plants in dry springs, which likely is related to significantly more precipitation in wet springs (56.2 mm and 80.6 mm in 2017 and 2018, respectively) than in dry springs (35.7 and 23.1 mm in 2019 and 2020, respectively). In addition, in an arid environment, climate variations may directly disrupt plant–pollinator interactions by modifying plant traits and affecting the community structure and behavior of floral visitors [47]. They may also indirectly influence these interactions by altering key food web linkages within communities, such as the ability of flower-dwelling predators to effectively capture floral visitors [48,49,50]. Therefore, precipitation influence may affect deserts pollinator diversity by reducing species richness and altering community composition. Meanwhile, due to *E. oxyrhinchum’s* cross-pollination of plants, precipitation influence also may change its flower attractiveness, including quality and quantity of nectar in flowers, with direct consequences for pollinator choice; thus, this may have negative effect on reproductive process of *E. oxyrhinchum* plants.

The number of seeds produced was higher for both SG and AG plants in wet springs than in dry springs, except for AG plants in the dry spring of 2019 (Figure 12). Adequate soil moisture for growth and reproduction in wet springs gives SG and AG plants a chance for high reproductive output. The seeds become part of the soil seed bank and were available for germination in the next good-precipitation year. However, in dry springs, the number of seeds that germinates and number of seeds SG plants produced decreased. In dry springs, many SG plants died due to low soil moisture (also see [36]), and plants were small with low seed production. On the other hand, if a wet autumn is followed by a dry spring, AG plants can use the limited soil moisture and produce a large number of seeds. For example, in the wet autumn of 2019, AG plants grew rapidly in early autumn and overwintered as a rosette. These AG plants quickly resumed growth in the dry spring of 2020 and had a competitive advantage over SG plants. In sum, then, it seems that biseasonal (autumn and spring) germination is a diversified bet-hedging strategy for *E. oxyrhinchum* in the precipitation-unpredictable cold desert environment of NW China.

Many more seeds with physical dormancy were produced by AG plants in wet than in dry springs (Figure 13). This is partly consistent with the concept of high reproduction-high dormancy, which avoids overcrowding in the next year and thus spreads germination risk over time [51,52], thereby avoiding the risk of population extinction in dry springs. SG plants had higher reproductive efficiency (biomass of reproductive organs) than AG plants in the dry spring, but reproductive output of individual plants of both SG and AG were similar in dry and wet springs. However, the percentage of dormant seeds of SG plants is flexible and inconsistent. The presence of seeds of *E. oxyrhinchum* with physical dormancy in the soil seed bank [37] helps ensure persistence of the population at a site, even if a dry autumn is followed by a dry spring.

In the SSP1 1.9 scenario, precipitation in spring and autumn will decrease slightly, but interannual precipitation will be highly variable. Thus, according to this scenario we predict that plants will produce seeds in years with wet springs, which will ensure the persistence of *E. oxyrhinchum* populations at sites with good and bad seed-production years. In the SSP2 4.5 and SSP3 8.5 scenarios, precipitation will increase in spring. Thus, we predict that SG plants will continue to dominate in the population. In addition, increased autumn precipitation will lead to an increase in number of AG plants present in spring. Thus, our study challenges the commonly-held perception based on correlative approaches (e.g., bioclimatic envelope approaches) suggesting that desert organisms may be particularly vulnerable to climate change [3,53]. Desert species such as *E. oxyrhinchum* that have evolved in harsh and unpredictable environments may be more resilient to increasingly erratic climates under global environmental change than those that evolved under predictable environmental conditions [3,54,55,56,57].

## 5. Conclusions

Given the flexibility in the life history of *E. oxyrhinchum* it seems unlikely that future climate change will have much of an impact on the life history and dominant position of this species in the herbaceous plant community in the unpredictable-rainfall environment of the Gurbantunggut Desert. In other words, it seems likely that *E. oxyrhinchum* will remain within the limits of its environment “comfort zone” as the climate continues to change in NW China. 

## Figures and Tables

**Figure 1 biology-10-00780-f001:**
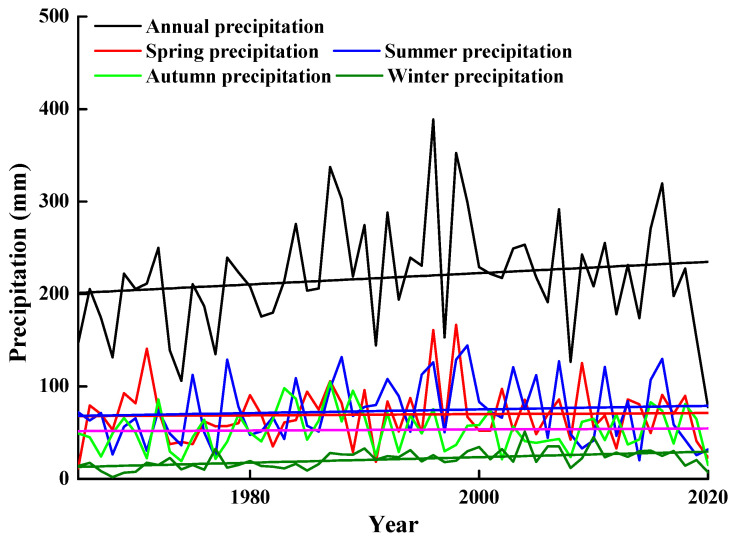
Precipitation at the Fukang weather station in spring, summer, autumn, and winter, 1965–2020.

**Figure 2 biology-10-00780-f002:**
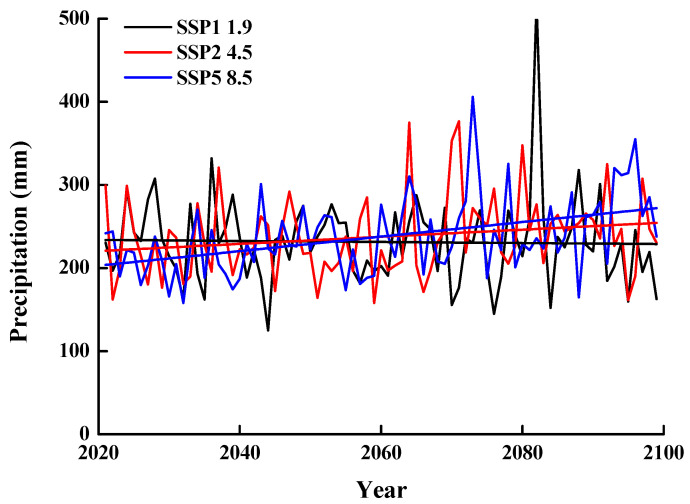
Projected annual precipitation in SSP1 1.9, SSP2 4.5, and SSP5 8.5 scenarios from 2021 to 2099, using data from the Fukang weather station.

**Figure 3 biology-10-00780-f003:**
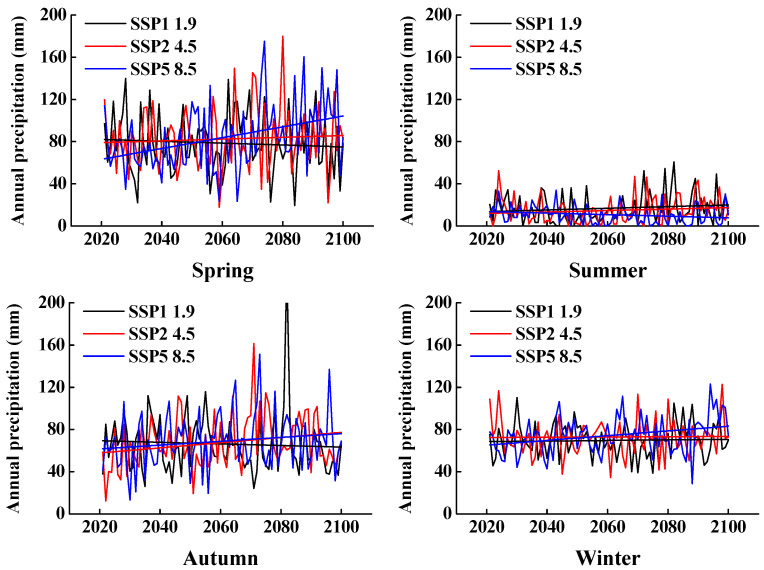
Projected precipitation in spring, summer, autumn, and winter in SSP1 1.9, SSP2 4.5 and SSP5 8.5 scenarios from 2021 to 2099, using data from the Fukang weather station.

**Figure 4 biology-10-00780-f004:**
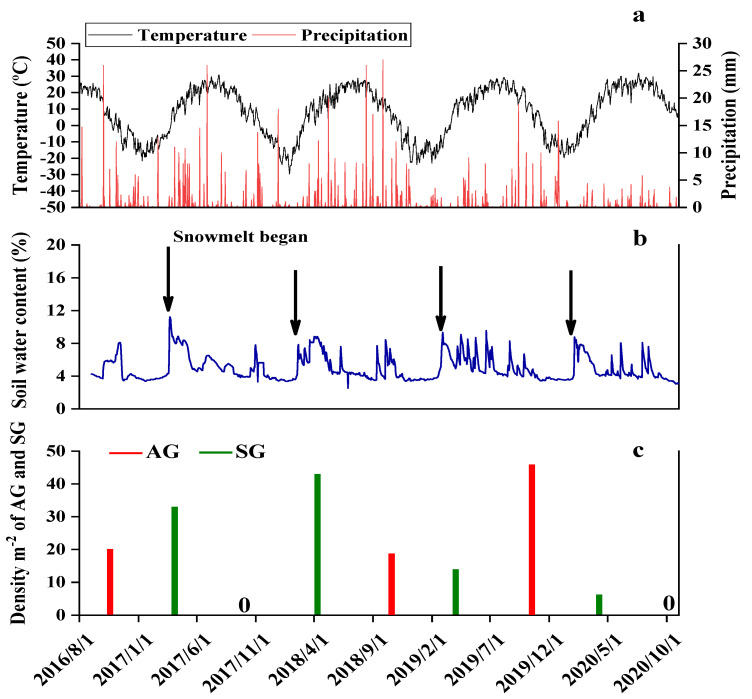
Soil temperature, precipitation (**a**), soil water content as a percentage of field weight at 5-cm-soil depth (**b**) and density of spring (SG)- and autumn (AG)-germinating plants of *Erodium oxyrhinchum* (**c**) from 1 September 2016 to 1 October 2020 at the study site.

**Figure 5 biology-10-00780-f005:**
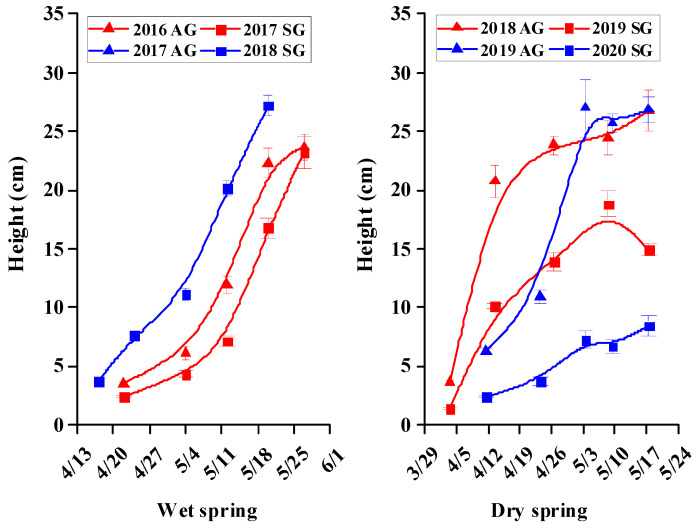
Increase in height of spring (SG)- and autumn (AG)-germinating plants of *Erodium oxyrhinchum* in wet (**left**) and dry (**right**). Vertical bars indicate standard errors of the means (n = 8).

**Figure 6 biology-10-00780-f006:**
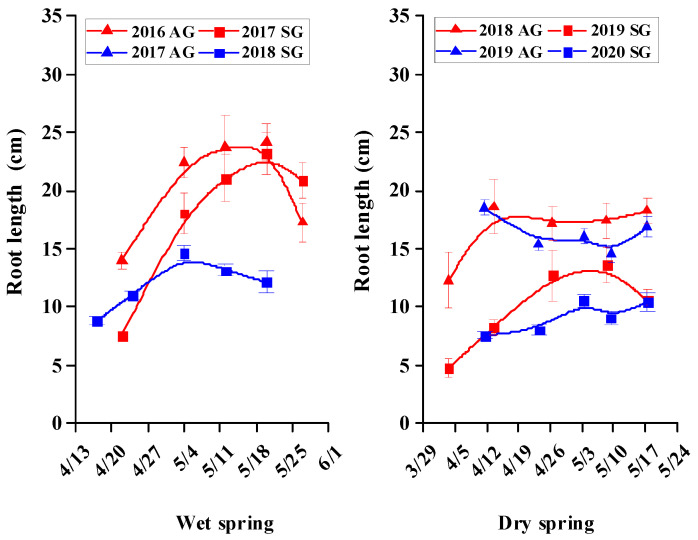
Increase in length of roots in spring (SG)- and autumn (AG)-germinating plants of *Erodium oxyrhinchum* in wet (**left**) and dry (**right**) springs. First three periods represent as vegetative stage, and last two periods represent as reproductive stage. Vertical bars indicate standard errors of the means (n = 8).

**Figure 7 biology-10-00780-f007:**
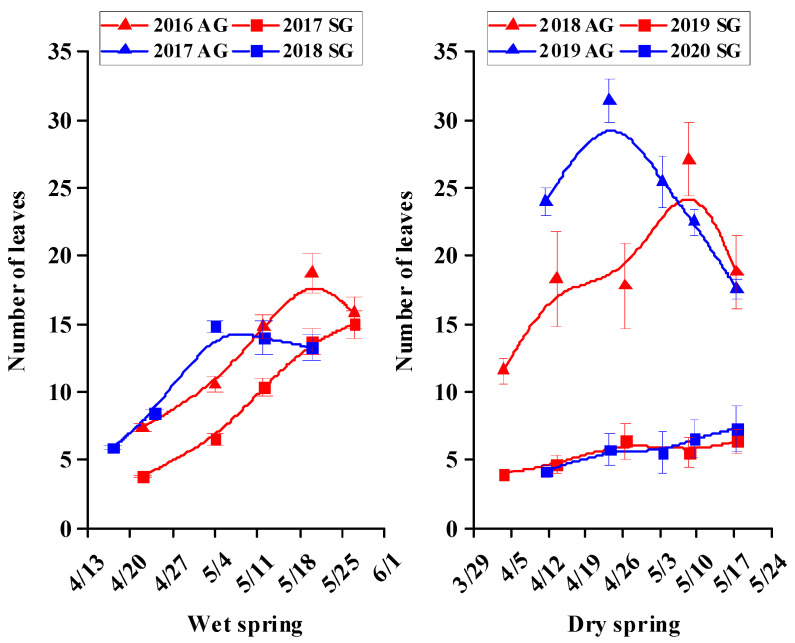
Increase in number of leaves of spring (SG)- and autumn (AG)-germinating plants of *Erodium oxyrhinchum* in wet (**left**) and dry (**right**) springs. The first three periods represent as vegetative stage, and the last two periods represent the reproductive stage. Vertical bars indicate standard errors of the means (n = 8).

**Figure 8 biology-10-00780-f008:**
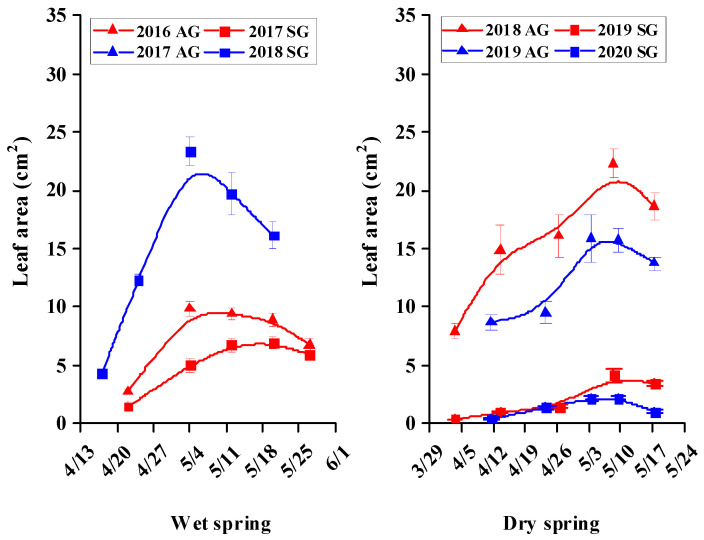
Increase in leaf area of spring (SG)- and autumn (AG)-germinating plants of *Erodium oxyrhinchum* in wet (**left**) and dry (**right**) springs. The first three periods represent the vegetative stage, and the last two periods represent the reproductive stage. Vertical bars indicate standard errors of the means (n = 8).

**Figure 9 biology-10-00780-f009:**
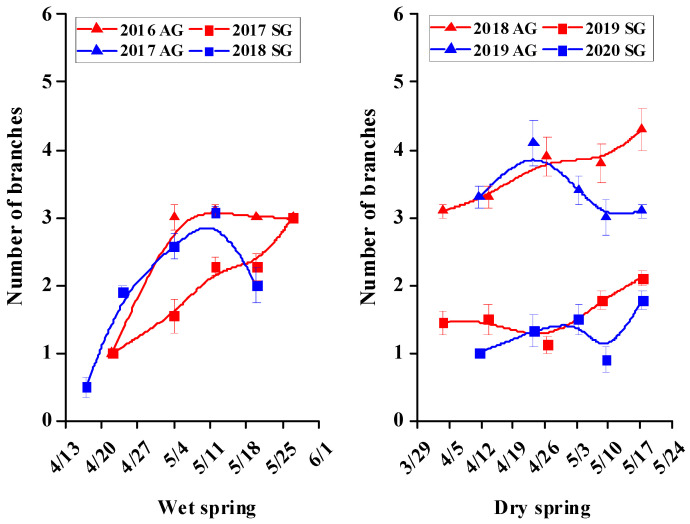
Increase in number of branches of spring (SG)- and autumn (AG)-germinating plants of *Erodium oxyrhinchum* in wet (**left**) and dry (**right**) springs. The first three periods represent the vegetative stage, and the last two periods represent the reproductive stage. Vertical bars indicate standard errors of the means (n = 8).

**Figure 10 biology-10-00780-f010:**
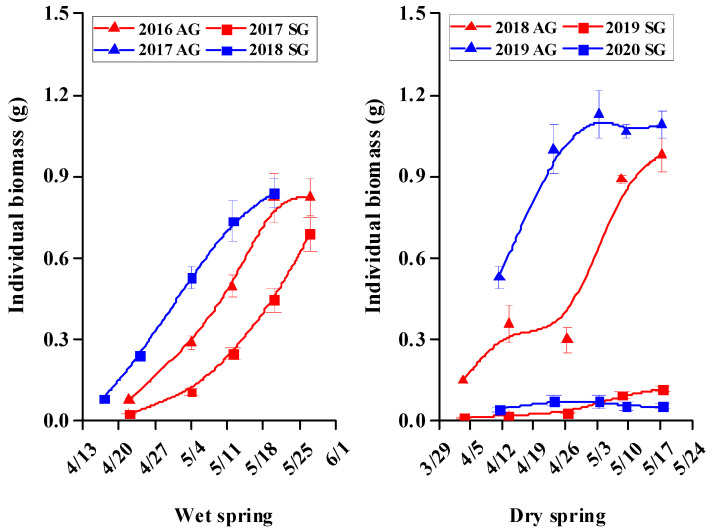
Increase in individual biomass of spring (SG)- and autumn (AG)- germinating plants of *Erodium oxyrhinchum* in wet (**left**) and dry (**right**) springs. The first three periods represent the vegetative stage, and the last two periods represent the reproductive stage. Vertical bars indicate standard errors of the means (n = 8).

**Figure 11 biology-10-00780-f011:**
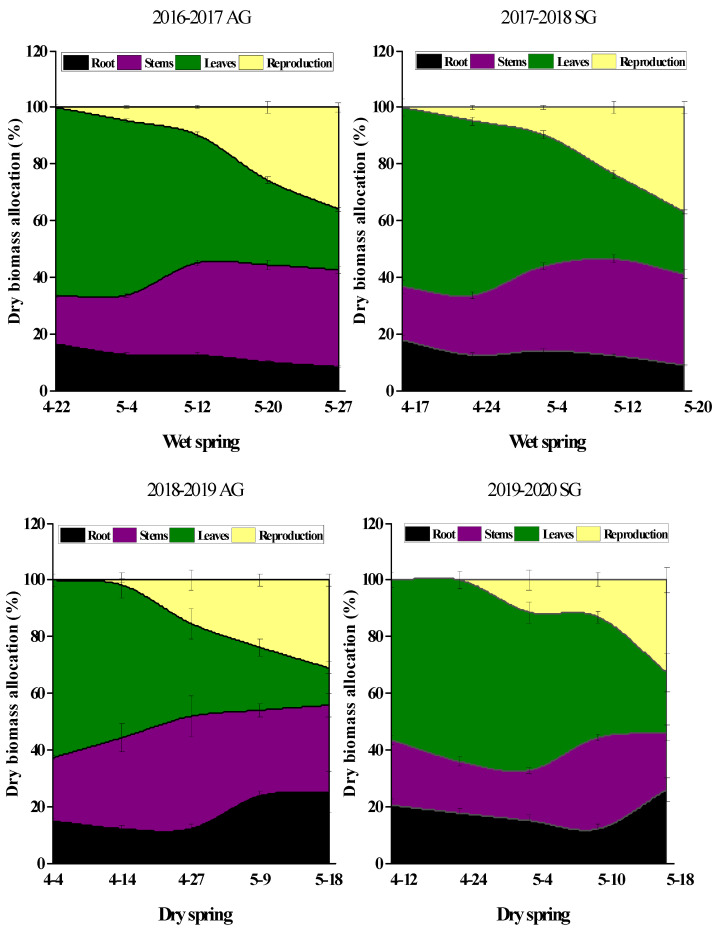
Changes in dry biomass allocation of spring (SG)- and autumn (AG)- germinating plants of *Erodium oxyrhinchum* in wet (**left**) and dry (**right**) springs. X axis represents different growth periods. The first three periods represent the vegetative stage, and the last two periods represent the reproductive stage. Vertical bars indicate standard errors of the means (n = 8).

**Figure 12 biology-10-00780-f012:**
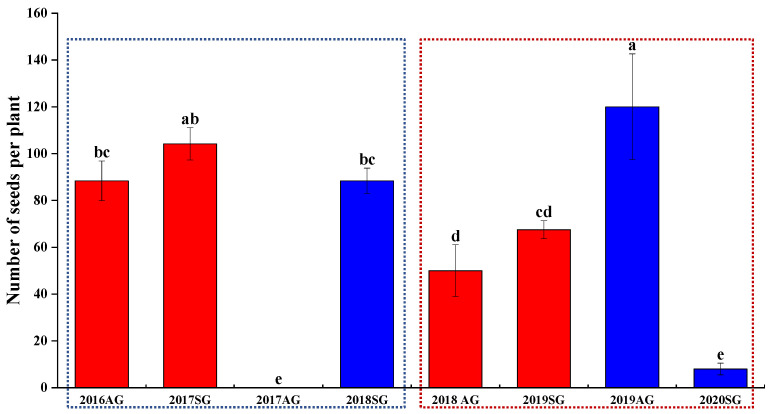
Number of offspring (F2) seeds per individual of spring (SG)- and autumn (AG)-germinating plants of *Erodium oxyrhinchum* in wet (**left**) and dry (**right**) springs. Different lowercase letters indicate significant differences (*p* < 0.05) among SG and AG in dry and wet springs. Vertical bars indicate standard errors of the means (n = 8).

**Figure 13 biology-10-00780-f013:**
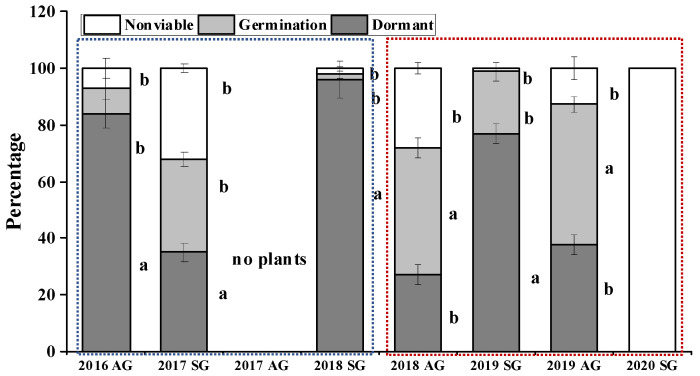
Nonviable, nondormant, and dormant seeds of offspring (F2) of spring (SG)- and autumn (AG)-germinating plants of *Erodium oxyrhinchum* in wet (**left**) and dry (**right**) springs. Different lowercase letters indicate significant differences (*p* < 0.05) among SG and AG in dry and wet springs. Vertical bars indicate standard errors of the means (n = 4).

## Data Availability

Data are contained within the article or Supplementary Material.

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
