# Peer review of "Is the Life History Flexibility of Cold Desert Annuals Broad Enough to Cope with Predicted Climate Change? The Case of Erodium oxyrhinchum in Central Asia"

_biology, 2021, doi:10.3390/biology10080780_

Round 1

Reviewer 1 Report

The manuscript reports a comprehensive study on the influence of future climate changes in a cold desert in NW China, considering different predictive climate models, on the life history and growth of an annual desert plant, Erodium oxyrhinchum. The research concludes that these foreseeable changes are unlikely to have a major impact on the life history of this plant and its prevalence in the herbaceous plant communities. However, there are some aspects to review:

-Some Latin names are written in italics and some are not.

- Orostachys spinosus should be replaced by Orostachys spinosa

Materials and Methods

-Seed germination in this study was tested at an alternating temperature, 25/10° C. It should be justified why this temperature was selected and what months in the natural habitat these temperatures are related to.

-The formula  GP= Germinated seeds (GS)/ Total number of seeds tested (TS). TS seeds are the sum of germinated seeds, viable seeds and nonviable seeds.

Does not corresponds to the formula of Chen et al. [12, 33] cited in the text:

“The final percentage of germination (FPG) was estimated as FPG = GN/SN, where GN is the total number of germinated seeds and SN is the number of viable seeds”

-Pag 7. Lines 248 and 250. m2 should be replaced by m2

Results

-The sections 3.5 to 3.7 are difficult to understand, it would be necessary to clarify them.

-Figures 5-9. It should be defined the duration of the “reproductive stage” for this plant in order to compare it with the vegetative stage.

-Figure caption 11 should be revised and the meaning of the x axis should be added

-The following sentences should be revised: (data differ or are not recorded in figure 13):

The percentage of nonviable F2 seeds produced by AG plants in wet springs was 7.0 (2017) and 2.0 (2018) and in dry spring 28.0 323 (2019) and 12.7 (2020)

However, dormancy of F2 seeds produced by AG plants in wet springs was 84.0% (2017) and 27.0% (2019) and in the dry spring 37.6% (2020).

However, germination of F2 seeds produced by AG plants in wet springs was 9.0% 329 (2017) and 45.0% (2019) and in the dry spring 49.7 % (2020) (Figure 13).

Discussion

The discussion should be improved. Several paragraphs correspond more to results than to discussion, it is convenient to reduce them in this section. The results of the experiment should be further contrasted with similar research and include more references. In this sense, some of the germination results need to be discussed in more depth

Author Response

Responses to the Reviewer 1’ comments

The manuscript reports a comprehensive study on the influence of future climate changes in a cold desert in NW China, considering different predictive climate models, on the life history and growth of an annual desert plant, Erodium oxyrhinchum. The research concludes that these foreseeable changes are unlikely to have a major impact on the life history of this plant and its prevalence in the herbaceous plant communities. However, there are some aspects to review:

-Some Latin names are written in italics and some are not.

Answer: done.

- Orostachys spinosus should be replaced by Orostachys spinosa

 Answer: done.

Materials and Methods

-Seed germination in this study was tested at an alternating temperature, 25/10° C. It should be justified why this temperature was selected and what months in the natural habitat these temperatures are related to.

Answer: We added a sentence ”This temperature regime simulates the mean daily maximum and minimum air temperatures in early spring in the Gurbantunggut Desert.”

-The formula  GP= Germinated seeds (GS)/ Total number of seeds tested (TS). TS seeds are the sum of germinated seeds, viable seeds and nonviable seeds.

Does not corresponds to the formula of Chen et al. [12, 33] cited in the text:

“The final percentage of germination (FPG) was estimated as FPG = GN/SN, where GN is the total number of germinated seeds and SN is the number of viable seeds”.

Answer: We changed it as Chen et al.’s published papers.

-Pag 7. Lines 248 and 250. m2 should be replaced by m2

Answer: done.

Results

-The sections 3.5 to 3.7 are difficult to understand, it would be necessary to clarify them.

Answer: We changed the figures, it can be easily to understand.

-Figures 5-9. It should be defined the duration of the “reproductive stage” for this plant in order to compare it with the vegetative stage.

Answer: First three periods represent as vegetative stage, and last two periods represent as reproductive stage.

-Figure caption 11 should be revised and the meaning of the x axis should be added

Answer: we added a sentence” X axis represents different growth periods.”

-The following sentences should be revised: (data differ or are not recorded in figure 13):

The percentage of nonviable F2 seeds produced by AG plants in wet springs was 7.0 (2017) and 2.0 (2018) and in dry spring 28.0 323 (2019) and 12.7 (2020)

However, dormancy of F2 seeds produced by AG plants in wet springs was 84.0% (2017) and 27.0% (2019) and in the dry spring 37.6% (2020).

However, germination of F2 seeds produced by AG plants in wet springs was 9.0% 329 (2017) and 45.0% (2019) and in the dry spring 49.7 % (2020) (Figure 13).

Answer: done.

Discussion

The discussion should be improved. Several paragraphs correspond more to results than to discussion, it is convenient to reduce them in this section. The results of the experiment should be further contrasted with similar research and include more references. In this sense, some of the germination results need to be discussed in more depth

Answer: We deleted results from discussion. And we added some references and compared with other studies. In addition, we discussed the relationship seed germination and seedling emergence with precipitation. Please see the details in the text.

Reviewer 2 Report

Dear Authors,

I am writing in reference to manuscript entitled " Is Life History Flexibility of Cold Desert Annuals Broad Enough to Cope with Predicted Climate Change? The Case of Erodium oxyrhinchum in Central Asia" by Huiliang Liu, Yanfeng Chen, Lingwei Zhang, Jerry M. Baskin, Carol C. Baskin, Lan Zhang, Yan Liu, Daoyuan Zhang and Yuanming Zhang.

The paper deals with the persistence (or not) of E. oxyrhinchum populations in the Central Asian cold desert into the future.

The introduction provides sufficient background and include relevant references. Materials and methods are described clearly and detailed enough. The results are clearly presented and the conclusions are supported by the results. Therefore, the paper is suitable for publication in its current form.

Best regards

Author Response

Responses to the Reviewer 2’ comments

I am writing in reference to manuscript entitled " Is Life History Flexibility of Cold Desert Annuals Broad Enough to Cope with Predicted Climate Change? The Case of Erodium oxyrhinchum in Central Asia" by Huiliang Liu, Yanfeng Chen, Lingwei Zhang, Jerry M. Baskin, Carol C. Baskin, Lan Zhang, Yan Liu, Daoyuan Zhang and Yuanming Zhang.

The paper deals with the persistence (or not) of E. oxyrhinchum populations in the Central Asian cold desert into the future.

The introduction provides sufficient background and include relevant references. Materials and methods are described clearly and detailed enough. The results are clearly presented and the conclusions are supported by the results. Therefore, the paper is suitable for publication in its current form.

Best regards

Answer: Thanks for recognizing our work.

Reviewer 3 Report

This paper describes a longitudinal (4-5 year) study on the effect of dry and wet springs/autumns on the germination and physiology of the annual desert plant, Erodium oxyrhinchum which is capable of 2 possible growing seasons.  The authors use data from climate change databases (CDS) and apply three different Socioeconomic models to predict future precipitation and temperature changes in the Gurbantunggut Desert in NW China.   Both spring and autumn-germinated plants were tracked over a 5 year time span.  Measurements of plant density, leaf area, root length, stem size, flower size, overall biomass, and F2 seed production and viability were taken.   Except for two seasons (autumns 2017 and 2020), plants continued to germinate regardless of wet or dry season.  However, wet or dry season did have an effect on biomass and plant growth. The overall goal of the paper was to apply climate change modesl to predict whether climate change will have an effect on this particular plant.  Based on the results, the authors main conclusion is that because this plant is adapted to living in an unpredictable climate, that climate change will not have a profound effect on its growth.   Although most of the methods in the paper are sound, there are some significant flaws in the design and in the manuscript that need to be addressed before it is publication-ready.  Here are my comments:

Introduction:

Lines 75-80:  Italicize all scientific names

Define "wet" vs. "dry" season.  Is there a cut off in precipitation that is outlined in the literature?

According to the literature, what is a reasonable time period that can be used as a longitudinal study to simulate climate change? Is four consecutive years long enough? Cite your reference. 

Does this plant self-fertilize or does it require a pollinator?

How long is the typical vernalization period for E. oxyrhinchum seeds?  

Methods:

You state in the methods that 16 plants "were marked" in the 9 subplots.  What does this mean?

What was your sample size (N) for harvesting (16?) for the plant measurements every 8 days for the season.  This is not included at all in the manuscript. 

Harvesting roots from dry, compacted soil can be difficult. How was this done so that the roots were not damaged and other plants in the same growth plots were not damaged? 

Lines 186-187 You describe using TTC as a method for determining viability for nongerminating seeds.  However, you don't describe this method or cite other studies that have used it. 

Results:  

Lines 252-254. These data for density would be better visualized as a bar graph.   Also, be clear that there were no germinated plants for the autumn 2017 and 2020 season.

Figures 3-10.  Why are there 2 different colors?  This does not seem to match/pair the treatment group so it is just confusing.   Also, you do not indicate what the error bars on the graphs represent or how many samples were used for each measurement (N).  Also there are no error bars for SG plants in Figure 10. 

Figure 11:  The naming conventions (ArO, ArT etc. are confusing).  Just using Allocation to: Flowers, Leaves, Stems, Root is clearer. 

Why were no correlation analyses done between temperature or precipitation and germination/morphological characters?  If you are suggesting that ppt had an effect on germination rates and other morphological characteristics, a relationship must be determined. 

Discussion/Conclusion:

My biggest concern with the manuscript is with the discussion.  Mostly, the discussion repeats the results and does not try to explain to the reader the importance or implications of the study.  Furthermore, the conclusions that because this plant will germinate in most seasons (wet or dry) and that predictions suggest that more rainfall will happen in this region the spring (ie. this plant will be ok because it is still in its "comfort zone" is grossly overstated.  

There are simply too many other ecological variables (competition with other plants, loss of pollinators, CO2 levels, etc. that may also affect the growth of this plant in the future) to make this broad statement. 

Furthermore, these studies are lacking controlled experiments where temperature and soil humidity are monitored and modulated over time. 

Due to a lack of correlation analyses, missing information about sample size and replication, and overstated conclusions, I am suggesting a major revision of the manuscript before it is published. 

Author Response

Responses to the Reviewer 3’ comments

This paper describes a longitudinal (4-5 year) study on the effect of dry and wet springs/autumns on the germination and physiology of the annual desert plant, Erodium oxyrhinchum which is capable of 2 possible growing seasons.  The authors use data from climate change databases (CDS) and apply three different Socioeconomic models to predict future precipitation and temperature changes in the Gurbantunggut Desert in NW China.   Both spring and autumn-germinated plants were tracked over a 5 year time span.  Measurements of plant density, leaf area, root length, stem size, flower size, overall biomass, and F2 seed production and viability were taken.   Except for two seasons (autumns 2017 and 2020), plants continued to germinate regardless of wet or dry season.  However, wet or dry season did have an effect on biomass and plant growth. The overall goal of the paper was to apply climate change modesl to predict whether climate change will have an effect on this particular plant.  Based on the results, the authors main conclusion is that because this plant is adapted to living in an unpredictable climate, that climate change will not have a profound effect on its growth.   Although most of the methods in the paper are sound, there are some significant flaws in the design and in the manuscript that need to be addressed before it is publication-ready.  Here are my comments:

Introduction:

Lines 75-80:  Italicize all scientific names

Answer: done.

Define "wet" vs. "dry" season.  Is there a cut off in precipitation that is outlined in the literature?

Answer: We added an introduction of Central Asia about precipitation in different years and seasons. It showed that precipitation occurred in all seasons, thus we defined wet vs. dry season as whether actual precipitation values are higher or lower than the mean precipitation values in the different seasons.

According to the literature, what is a reasonable time period that can be used as a longitudinal study to simulate climate change? Is four consecutive years long enough? Cite your reference. 

Answer: Thanks for your suggestion. We added reference in the discussion. Please the detail in the text.

Does this plant self-fertilize or does it require a pollinator?

Answer: We added “It's cross-pollination, relying on insects like bees.” in the introduction. Please see the details in the text.

How long is the typical vernalization period for E. oxyrhinchum seeds?  

Answer: We added “E. oxyrhinchum seeds have physical dormancy, and dormancy release needed high temperature in summer and low temperature in winter. “in the introduction. Please see the details in the text.

Methods:

You state in the methods that 16 plants "were marked" in the 9 subplots.  What does this mean?

Answer: 16 plants were random marked in the subplots. We changed it.

What was your sample size (N) for harvesting (16?) for the plant measurements every 8 days for the season.  This is not included at all in the manuscript. 

Answer: For SG and AG plants, due to the relatively small plant biomass at the seedling stage, but plant biomass reached about 1g at mature stage. Thus, 10 seedlings or 8 mature individuals were harvested for the plant measurements at the seedling stage and mature stage, respectively, and the sample size (N) of each treatment every 8 days is 8~10.

Harvesting roots from dry, compacted soil can be difficult. How was this done so that the roots were not damaged and other plants in the same growth plots were not damaged? 

Answer: Because E. oxyrhinchum is tap root plant, so its root can obtain easily from the soil.

Lines 186-187 You describe using TTC as a method for determining viability for nongerminating seeds.  However, you don't describe this method or cite other studies that have used it. 

 Answer: We added a cite paper.

Results:  

Lines 252-254. These data for density would be better visualized as a bar graph.   Also, be clear that there were no germinated plants for the autumn 2017 and 2020 season.

Answer: We changed it.

Figures 3-10.  Why are there 2 different colors?  This does not seem to match/pair the treatment group so it is just confusing.   Also, you do not indicate what the error bars on the graphs represent or how many samples were used for each measurement (N).  Also there are no error bars for SG plants in Figure 10. 

Answer: We changed it as your suggestions. Please see the details in the text.

Figure 11:  The naming conventions (ArO, ArT etc. are confusing).  Just using Allocation to: Flowers, Leaves, Stems, Root is clearer. 

Answer: Done.

Why were no correlation analyses done between temperature or precipitation and germination/morphological characters?  If you are suggesting that ppt had an effect on germination rates and other morphological characteristics, a relationship must be determined. 

Answer: The correlation between temperature or precipitation and germination did exist during certain periods of the plant life history, and the correlation between temperature or precipitation and germination should be different in different life history stages. For example, there was a one-to-one correspondence between the seedling emergence of AG plants and precipitation in autumn (from the end of September to the end of October), and the greater the precipitation, the higher the emergence percentage. But when precipitation occurred before the end of September, or after the end of October, too high or too low temperature often inhibited seedlings emergence. Similarly, for SG plant, which germinated in early spring using snow melt water, thus, the seedlings emergence of SG plants corresponds to the melting time of snow in early spring. Meanwhile, the melting time of snow was mainly affected by temperature, and the amount of melted water was affected by the thickness of snow. Therefore, temperature is the main environmental factor driving the emergence of SG plants, but precipitation has an important influence on the emergence of AG plants. This description is in the text.

Discussion/Conclusion:

My biggest concern with the manuscript is with the discussion.  Mostly, the discussion repeats the results and does not try to explain to the reader the importance or implications of the study.  Furthermore, the conclusions that because this plant will germinate in most seasons (wet or dry) and that predictions suggest that more rainfall will happen in this region the spring (ie. this plant will be ok because it is still in its "comfort zone" is grossly overstated.  

Answer: We deleted this sentence in the conclusion.

There are simply too many other ecological variables (competition with other plants, loss of pollinators, CO2 levels, etc. that may also affect the growth of this plant in the future) to make this broad statement. 

Answer: We added some discussions about precipitation affecting pollinators, due to E. oxyrhinchum is cross-pollination plant. Please see the details in the text.

Furthermore, these studies are lacking controlled experiments where temperature and soil humidity are monitored and modulated over time. 

Answer: In the methods, on 1 September 2016, we installed a weather monitoring device (Caipos GmbH, 149 Schillerstrasse Gleisdorf, Austria) at the study site to measure amount of precipitation and soil temperature and soil moisture content, which were recorded hourly at a depth of 5 cm.

Due to a lack of correlation analyses, missing information about sample size and replication, and overstated conclusions, I am suggesting a major revision of the manuscript before it is published. 

Answer: Descriptions is as above

Round 2

Reviewer 3 Report

The authors addressed most of my concerns and issues with this paper.  However, the conclusions are still grossly overstated.  The entire conclusion section can and should be deleted.  

Furthermore, the authors did not indicate in the graphs (or in the methods) what error bars represent.  Figure captions should also include sample size (N).  This is standard scientific practice. 

Author Response

We deleted absolutely the discussion. We added the description about bar in the figure caption, it represt indicate standard errors of the means (n = 8).